# An Investigation of Employment Hope as a Key Factor Influencing Perceptions of Subjective Recovery among Adults with Serious Mental Illness Seeking Community Work

**DOI:** 10.3390/bs14030246

**Published:** 2024-03-19

**Authors:** Marina Kukla, Alan B. McGuire, Kenneth C. Weber, Jessi Hatfield, Nancy Henry, Eric Kulesza, Angela L. Rollins

**Affiliations:** 1HSR&D Center for Health Information and Communication, Richard L. Roudebush VA Medical Center, Indianapolis, IN 46202, USA; alan.mcguire@va.gov (A.B.M.); nancy.henry2@va.gov (N.H.); angela.rollins@va.gov (A.L.R.); 2Department of Psychology, Indiana University-Purdue University Indianapolis, Indianapolis, IN 46202, USA; 3Edward Hines Jr. VA Hospital, Hines, IL 60141, USA; kenneth.weber@va.gov (K.C.W.); eric.kulesza@va.gov (E.K.); 4St. Louis VA Health Care System, St. Louis, MO 63103, USA; jessica.hatfield@va.gov; 5Center for Health Services Research, Regenstrief Inc., Indianapolis, IN 46202, USA

**Keywords:** vocational rehabilitation, recovery, serious mental illness, employment hope, veterans

## Abstract

Introduction: Employment is an important contributor to recovery in people with serious mental illness (SMI), yet studies have not explored how subjective elements of employment hope contribute to perceptions of global recovery in this population. Methods: The current study examined the relationship between employment hope and subjective recovery in 276 unemployed adults with SMI participating in a multi-site clinical trial of a cognitive behavioral group intervention tailored toward work and combined with vocational rehabilitation. Participants had diagnoses of schizophrenia spectrum, bipolar, depressive, and posttraumatic stress disorders, and were receiving services at three Veterans Affairs healthcare facilities in the United States. Data were collected at study baseline. Linear regression analysis examined the relationship between employment hope (Short Employment Hope Scale; EHS-14) and subjective recovery (Recovery Assessment Scale; RAS) after controlling for psychiatric symptom severity and mental-health-related burden on daily life. Results: After accounting for covariates, employment hope significantly contributed to the regression model explaining subjective recovery. The overall model of predictor variables explained 52.5% of the variance in recovery. The results further explore the relationships between EHS-14 and RAS subscales. Conclusions: The findings suggest that employment hope is a key intervention target to bolster subjective recovery in this vulnerable population.

## 1. Introduction

Recovery in serious mental illness (SMI) has been extensively examined in recent decades, highlighting both objective and subjective components that emerge over time in a majority of people [1]. The study of recovery from SMI is particularly important, as it has been found to serve as a protective factor in the face of more severe illness, preserving quality of life across cognitive, affective, and participation domains (e.g., [2]). Current conceptualizations of recovery in this population emphasize person-centered elements, including autonomy, connection with others, attaining purpose, self-experience and identity, and increased empowerment [3,4]. Consistent with these definitions is the notion that as individuals with SMI progress through recovery, they engage more fully in the world, make meaning as active agents, gain skills, and form an increasingly integrated sense of themselves in the world, how they want to face their challenges, and the goals they wish to pursue (e.g., [1,5,6,7]).

These comprehensive views of subjective recovery in SMI can be linked with objective recovery elements, such as participation in the community, effectively engaging with others, involvement with hobbies and interests, educational attainment, and notably, taking part in employment (e.g., [8]). Though a range of environments exist in which people with SMI engage in gainful work, findings support participation in competitive jobs, positions open to anyone with or without a disability, as particularly powerful to promote recovery (e.g., [8]). Accordingly, studies have found that people with SMI who are engaged in competitive employment report higher subjective levels of recovery, have better well-being and life satisfaction, make improvements in self-esteem, and have more favorable clinical indicators of recovery, such as fewer days of psychiatric hospitalization and lower levels of psychiatric symptoms across illness domains [9,10]. 

Similarly, military veterans with serious mental illness also struggle in attaining and keeping community jobs; some estimates demonstrate that three-quarters of this population are unemployed [11]. Though they are a distinct population compared with non-veterans, they also experience poorer well-being across a number of domains as a result of long-term unemployment and inconsistent employment. For instance, unemployment in veterans with SMI is linked with greater poverty, more reliance on entitlements and the service system [12], and more trouble meeting basic needs [13]. Studies have also demonstrated that among veterans with SMI, lack of participation in employment has major economic and well-being effects that can persist into the future if not addressed [14]. Contrastingly, studies of veterans with SMI and broader meta-analysis evidence indicates that work leads to major gains across health, quality of life, and recovery domains [15]. Furthermore, a sense of purpose, such as that found through employment, was negatively associated with suicidal thoughts in veterans with posttraumatic stress disorder (considered a serious mental illness by the Department of Veteran Affairs [16].

While there are clear literature reports linking employment participation with stronger recovery in both veteran and non-veteran populations with SMI, there is a dearth of studies examining subjective elements of employment and how these relate to person-centered perspectives of global recovery in any SMI group. In particular, there is evidence that employment can lead to a re-emergence of individual meaning and purpose; however, the psychological components involved or prerequisites of this process have yet to be identified and studied. A recently defined construct, employment hope, describes a state in which individuals possess the internal strength to face challenges required to attain work and the sense of self geared toward vocation; employment hope comprises work-specific self-determination, agency, resilience, and empowerment [17,18]. Employment hope is also characterized as a transformative process in which a person becomes empowered and motivated for the future, envisioning the possibility of meaningful and satisfying employment, progresses toward work goals using resources and skills [19]. Studies of employment hope have found that it is directly related to more positive work outcomes [20], and more recently, it has been found to mediate the relationship between job-related barriers and economic self-sufficiency in people with mental illness through the pathway of increasing psychological self-sufficiency [21]. Further, among a sample of people formerly involved with the criminal justice system, employment hope mediated the path between self-esteem and economic self-sufficiency [22]. Taken together, employment hope is a phenomenon that may link the desire to work with successful work outcomes and the attainment of recovery through the pathways of agency, mastery, meaning, and purpose. 

Therefore, in the current investigation, it is expected that employment hope will be significantly associated with overall subjective recovery and particularly with the recovery domains related to a goal and success focus and self-confidence. Relationships between employment hope subdomains and recovery subdomains will also be exploratorily examined.

## 2. Methods

### 2.1. Overview

The current study is a cross-sectional, secondary analysis of baseline data from a multi-site, two-armed randomized controlled trial (RCT). The RCT parent study examined the work, employment hope, health, and recovery effects of a cognitive behavioral group intervention tailored to enhance competitive work outcomes, Cognitive Behavioral Therapy for Work Success (CBTw), in unemployed adults with serious mental illness. Data regarding the current secondary analysis were collected prior to study randomization of the CBTw clinical trial or receipt of study interventions (CBTw and psychoeducation control interventions).

### 2.2. Participants

Participants were recruited from three sites at VA Medical Center vocational rehabilitation programs in three midwestern US states. Veterans aged 18 and older receiving Veterans Health Administration vocational services promoting competitive work were eligible for the parent study if they (1) were unemployed; (2) had a medical record diagnosis of a serious mental illness, including schizophrenia spectrum disorders, bipolar disorders, depressive disorders, or posttraumatic stress disorder (PTSD); and (3) had a competitive work goal in the community. Exclusion criteria were (1) previous participation in a cognitive behavioral therapy intervention tailored for competitive work (CBTw); (2) a severe medical or cognitive disorder (e.g., dementia) that would prevent participation in study activities. All vocational rehabilitation services were focused on assisting veterans to secure competitive work in the community; veterans were at various stages of VR participation. Further, vocational rehabilitation professionals at each site referred potentially eligible participants. Potential participants were also identified via electronic medical record review of vocational services encounters, mailed a study informational sheet, and subsequently contacted by research staff who provided more information about the study.

### 2.3. Procedures 

Participants were recruited at the three sites in six waves spanning from March 2021 to January 2023. Participants provided verbal informed consent upon study enrollment. At baseline, trained study staff conducted the employment and psychosocial assessments virtually (via telephone or video). Participants received $20 for completion of the baseline assessments. All study procedures were approved by the VA Central Institutional Review Board and local VA regulatory boards at all sites. As discussed below, psychometric testing was performed for the primary independent and dependent variables, employment hope, and subjective recovery.

### 2.4. Measures 

The Recovery Assessment Scale (RAS) is a 41-item, self-reported measure of subjective recovery attitudes. The RAS is scored on a 1-to-4 Likert scale with higher scores indicating stronger recovery attitudes. The measure has five subscales including Confidence and Hope; Willingness to Ask for Help; Goal and Success Orientation; Reliance on Others; and No Domination by Symptoms. The RAS was developed and assessed in adults with SMI, demonstrating strong validity and reliability across studies [23]. Because this measure was developed for people with SMI and the psychometric properties have been extensively assessed in this specific SMI population, further examination of reliability and validity was not conducted in this study. Employment hope was assessed by the Short Employment Hope Scale (EHS-14), producing an overall mean and two subscales: Psychological Empowerment and Goal-Oriented Pathways [20]. The EHS-14 is scored on a 1-to-10 Likert scale with higher scores representing stronger levels of employment hope. The EHS-14 has shown strong reliability and validity in samples that include people with mental illness [21]. In this study, Cronbach’s alpha was 0.925 for the full EHS-14 scale (all items), 0.862 for the Psychological Empowerment subscale and 0.895 for the Goal-Oriented Pathways subscale. Overall, this indicates very strong internal consistency. Further psychometric testing (e.g., average variance extracted; composite reliability) was not conducted due to significant factory analyses conducted in prior studies and the strong internal consistency found here. Psychiatric symptoms were examined using the adult version of the DSM-5 Level 1 Cross-Cutting Symptom Measure (CCSM), a 23-item, Likert-response self-reporting tool that is scored on a scale of 0 to 4 with higher scores indicating more severe psychiatric symptoms. The CCSM includes 13 symptom domains relevant to symptoms across serious mental illness diagnoses including depression, anger, mania, anxiety, somatic, suicidal ideation, psychosis, sleep problems, memory, repetitive thoughts and behaviors, dissociation, personality functioning, and substance use. In this study, CCSM total scores were used to gauge overall symptom burden because the sample was diagnostically diverse. The CCSM has been found to have strong validity and reliability in people with SMI [24]. The Veterans Rand 12-item health survey (VR-12) was used to assess health burden on daily life. The VR-12 is comprised of the Mental Component Summary (MCS) and the Physical Component Summary [25]. The MCS and PCS are standardized with a mean of 50 and a standard deviation of 10, with higher scores indicating a lower health burden and higher health-related quality of life. The MCS and PCS have good reliability and are predictive of health outcomes in veterans with chronic health conditions.

### 2.5. Data Analysis

All analyses were conducted using SPSS 29. First, preliminary analyses were conducted to assess adherence to statistical assumptions. Next, descriptive statistics were used to characterize the study sample and the baseline outcomes and covariates, including recovery attitudes, employment hope, health-related quality of life, and psychiatric symptoms. Third, linear multiple regression analyses were run using stepwise entry, controlling for overall psychiatric symptoms and VR-12 Mental Component scores in the first block. Then, the independent variable, employment hope scores were entered in the second block and were regressed on overall recovery scores, RAS total. This stepwise linear regression approach was used in order to account for covariates that may influence subjective recovery. For instance, subjective recovery could be suppressed by high levels of psychiatric symptoms or a significant burden of symptoms on daily life. This statistical approach removes those possibilities and better hones in on the association between subjective employment hope and global recovery. Finally, bivariate correlations were run to examine the relationship between RAS subscales and EHS-14 subscales. Significance levels were set at *p* < 0.01 to account for multiple comparisons.

## 3. Results

### 3.1. Background Characteristics

Complete background characteristics are presented in a manuscript that is under review elsewhere. The total sample comprised 276 participants with 93 at both sites 1 and 2 and 90 at site 3. Most participants identified as male (78%) and 22% identified as female or transgender female. Thirty-seven percent identified as Black or African American and 53% identified as Caucasian or white. The mean age at the study baseline was 48.5 years (SD = 12.3) and primary psychiatric diagnoses included schizophrenia spectrum and psychotic disorders (N = 24); bipolar disorders (N = 44); depressive disorders (N = 116); and posttraumatic stress disorder (N = 92).

### 3.2. Preliminary Analyses and Baseline Findings

All assumptions of statistical tests were met. The data were normally distributed. Secondly, the relationship between independent variables and dependent variables was linear. Thirdly, homoscedasticity of the variance of the residuals was preserved, as indicated by the standard practice of examination of residual scatterplots [26]. In addition, of note, multicollinearity among the independent variables was not an issue after examining the variance inflation factor and tolerance statistics. In particular, for the final regression model, the variance inflation factor had a value of 1.88 for the VR-12 Mental Component Scale, 1.71 for CCSM total symptom scores, and 1.27 for EHS-14 overall mean scores. Regarding tolerance values, VR-12 Mental Component Scores had a value of 0.532, CCSM total scores had a value of 0.585, and EHS-14 overall mean scores had a value of 0.787. For the variance inflation factor, these values fell well below the threshold of 10 indicating a possible issue with multicollinearity; for tolerance, these values fell well above the threshold of 0.1, indicating potential problems with multicollinearity [27]. Descriptive statistics for overall employment hope, overall subjective recovery, and EHS-14 and RAS subscales are shown in Table 1. The VR-12 Physical Component Scale was not significantly related to RAS or EHS-14 scores and was therefore not included in the main analyses presented below.

### 3.3. Linear Regression Analyses

The final regression model including CSSM symptom total scores, VR-12 MCS scores, and EHS-14 mean scores explained 52.5% of the variance in RAS total scores. By adding in CCSM total scores and VR-12 MCS scores in the first block, symptom levels and impacts on daily life were controlled for before adding in employment hope into the second block. Specifically, after adding EHS-14 mean scores in the second block of the model, the change was significant, F change (1, 271) = 122.11, *p* < 0.001. In the final model, F(3, 271) = 99.72, *p* < 0.001, and EHS-14 total scores (beta = 0.522, *p* < 0.001), MCS scores (beta = 0.159, *p* = 0.006), and CCSM total scores (beta = −0.198, *p* < 0.001) were significant.

### 3.4. Employment Hope and Recovery Subscale Correlational Analyses

As is shown in Table 2, the EHS-14 Goal-Oriented Pathways subscale was most strongly related to RAS total scores, *p* < 0.001. The EHS-14 subscale, Psychological Empowerment, was also significantly related to overall global recovery, *p* < 0.001. 

Second, EHS-14 Psychological Empowerment was significantly related to all RAS subscales (*p* < 0.001), with particularly strong relationships with RAS Confidence and Hope (r = 0.57) and RAS Goal and Success Orientation (r = 0.55). Small effect sizes characterized the association between EHS-14 Psychological Empowerment with RAS Reliance on Others; small to medium relationships were found between Psychological Empowerment with RAS subscales Willingness to Ask for Help and No Domination by Symptoms. 

Third, the EHS-14 Goal-Oriented Pathways subscale was significantly related to all RAS subscales (*p* < 0.001) and similarly, was most strongly associated with RAS Confidence and Hope (r = 0.57) and RAS Goal and Success Orientation (r = 0.56). EHS-14 Goal-Oriented Pathways had small to moderate correlations with RAS No Dominations by Symptoms and Willingness to Ask for Help. The subscale, Goal-Oriented Pathways. was modestly associated with RAS Reliance on Others.

## 4. Discussion

This study examined the relationship between employment hope, a novel construct comprising cognitive and non-cognitive psychological processes by which individuals move toward work goals, and global recovery in unemployed adults with serious mental illness who were enrolled in a multi-site randomized controlled trial testing group-based cognitive behavioral therapy tailored for competitive work. The current investigation found that employment hope was significantly associated with overall perceptions of recovery and all recovery sub-domains after controlling for psychiatric symptoms and mental health burden on daily life. As hypothesized, employment hope was most strongly related to the recovery constructs reflecting confidence and hope and orientation toward personal goals. These are the first published findings to demonstrate that people with SMI who possess more positive perceptions of themselves and firm ideas about the paths they wish to pursue have more optimistic views of their ability to successfully pursue employment. These findings are consistent with the possibility that recovery aspects of personal agency (e.g., RAS item 4: “I believe I can meet my current personal goals”), mastery (e.g., RAS item 2: “I have my own plan for how to stay or become well”), and coherent sense of self (e.g., RAS item 20: “I have an idea of who I want to become”) are associated with key psychological underpinnings of work success. Further, it may be that this is a bidirectional relationship in which subjective recovery and employment hope contribute to one another and result in better personal outcomes; another possibility may be that both are related or stem from a third unmeasured variable. Or, it may be that strong subjective recovery is a prerequisite for hope in the targeted area of work, or rather, that feeling hopeful about one’s employment prospects and ability to succeed at work generalizes to influence broader recovery attitudes. Future research should examine these possibilities in people with SMI to optimally target those phenomena with rehabilitation and psychotherapy interventions.

Notably, both employment hope subscales, Psychological Empowerment and Goal-Oriented Pathways, were associated with global recovery and all recovery subdomains. Particularly, consistent with overall recovery scores, Psychological Empowerment and Goal-Oriented Pathways were related most strongly to RAS Confidence and Hope and Goal and Success Orientation, characterized by moderate effect sizes. Employment hope subscales were only modestly related to other RAS domains, perhaps unexpectedly. For instance, it may have been anticipated that EHS-14 items related to utilization of skills and resources (as a part of the Goal-Oriented Pathways subscale) would be more strongly associated with RAS subdomains reflecting the use of support from others and willingness to ask for help in line with recovery goals. Rather, it may be that using tangible resources to work toward job-related goals is distinct from relying on natural supports in a more complex interpersonal landscape that people with SMI must navigate in their daily lives as they pursue recovery. 

Further study should examine the nuances of employment hope as a psychological underpinning of both objective work success and as a contributor to global recovery in SMI. This is particularly important as employment hope has been adopted as a measure to administer over time and inform treatment planning decisions in many Veterans Health Administration VR programs that serve individuals with SMI. Accordingly, understanding how employment hope changes over time along with subjective recovery is another key area of future study. In this vein, real-world clinical evidence suggests that both employment hope and subjective recovery are high as vocational services begin before the realities of finding and securing employment set in; it should be noted that participants were at all stages of employment services during the current study, controlling for this potential issue. As the common barriers and issues regarding attaining employment are realized, employment hope may act as more of a buffer toward potential losses in subjective recovery. This notion is consistent with findings that demonstrate that the metacognitive construct of mastery protects against lower levels of subjective recovery through the pathway of functional skills competence [28]. In other words, those with better internal tools and more effective day-to-day self-management [29] may have higher levels of employment hope, which may preserve a sense of recovery in the face of challenges; this may ultimately lead to better outcomes, such as the achievement of one’s goals in the community, including those related to work. Much future research is needed to examine these hypotheses in order to determine causal pathways and best tailor interventions to support people with SMI as they strive toward their community goals while maintaining subjective recovery and well-being.

In total, study findings suggest that employment hope may be an independent intervention target to bolster recovery in people with SMI. One intervention, cognitive behavioral therapy tailored for competitive work (CBTw), incorporates content designed to bolster job-related self-confidence, sense of self surrounding employment, and instill hope that work success is possible [30]. Previous findings indicate that it can promote recovery attitudes, increase work-related self-efficacy, and reduce symptoms [31]. In concert with CBTw, vocational rehabilitation approaches assist individuals with SMI to achieve their personal work goals in a manner and setting of their choosing [32]. Attaining one’s work objectives and successfully being more active in the community contributes to recovery (e.g., [9,10]) and may reciprocally boost employment hope. Another strengths-based approach designed to enhance recovery in people with serious mental illness facilitates the integration of information about oneself, others, and the community, and the use of that information to identify strategies to deal with challenges and pursue personal goals [33]. These metacognitively oriented interventions and associated improvements in metacognitive capacities have been found to promote a coherent sense of self and a range of positive outcomes, including subjective and objective recovery in people with SMI [7,34]. Of note, metacognition is predictive of future community job acquisition in people with SMI [35]. Taken together, these approaches hold promise to impact perceptions of employment hope via multiple pathways. As such, studying employment hope among people with SMI who are not actively receiving vocational supports but may be pre-contemplative or contemplative may be crucial and supported by these recovery oriented psychotherapy approaches.

This study has limitations that are worthy of note. In particular, the data are cross-sectional and causality between recovery and employment hope cannot be determined. Future research should seek to understand these relationships using designs that can better determine causal relationships. In addition, the study measures were all self-reported, and objective indicators of outcomes (e.g., work outcomes data) are lacking at this baseline time point. However, subjective input on recovery and employment hope are necessary to understand these person-centered phenomena and the construct of identity at the core of recovery in SMI. Further, most participants identified as middle-aged males, of Black or Caucasian/White race and ethnicity, and all had a military background; participants were from three midwestern locations and all received services from Department of Veteran Affairs settings. These findings regarding employment hope and global recovery may not be generalizable to other groups of people with SMI outside of the veteran and VA user population or to those in variable racial, ethnic, age, and geographical groups.

In conclusion, employment hope—a cognitive, emotional, and motivational construct in which one possesses the belief that purposeful work is attainable—is positively associated with global recovery in adults with serious mental illness, after accounting for the impact of psychiatric symptoms and mental health challenges on daily life. These results place employment hope as a novel target through which recovery may be directly fostered via a range of evidence-based rehabilitative and psychosocial approaches.

## Figures and Tables

**Table 1 behavsci-14-00246-t001:** Recovery and employment hope outcome descriptives.

Variable	N	M	SD
EHS-14 overall mean	275	7.59	1.67
EHS-14 subscale 1, Psychological Empowerment	275	7.83	1.77
EHS-14 subscale 2, Goal-Oriented Pathways	275	7.42	1.80
RAS total score at BL	276	161.53	19.42
RAS subscale 1, Confidence and Hope	276	3.84	0.66
RAS subscale 2, Willingness to Ask for Help	276	3.96	0.81
RAS subscale 3, Goal and Success Orientation	276	4.24	0.60
RAS subscale 4, Reliance on Others	276	3.96	0.74
RAS subscale 5, No Domination by Symptoms	276	3.01	1.01

**Table 2 behavsci-14-00246-t002:** Recovery and employment hope correlation matrix.

	RAS Total	RAS Confidence and Hope	RAS Willingness Ask for Help	RAS Goal and Success Orient.	RAS Reliance on Others	RAS No Dom. Symp.
EHS-14 Overall Mean	0.664 **	0.610 **	0.392 **	0.598 **	0.244 **	0.409 **
EHS-14 Psy. Empower.	0.604 **	0.567 **	0.340 **	0.549 **	0.220 **	0.349 **
EHS-14 Goal Pathways	0.630 **	0.571 **	0.384 **	0.564 **	0.234 **	0.405 **

** Correlation is significant at the 0.01 level (two-tailed).

## Data Availability

De-identified data reported in this paper and study materials may be available upon request.

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
