# Peer review of "An Investigation of Employment Hope as a Key Factor Influencing Perceptions of Subjective Recovery among Adults with Serious Mental Illness Seeking Community Work"

_behavsci, 2024, doi:10.3390/bs14030246_

Round 1

Reviewer 1 Report

Comments and Suggestions for Authors

The only concern I have is the cross sectional nature of the dataset may not interpret as causal relationship.

Author Response

Reviewer 1:

  1. The only concern I have is the cross sectional nature of the dataset may not interpret as causal relationship.

Response: We have further expanded our discussion of this limitation in the limitation section of the Discussion and included causal designs as an important area of future research that might better delineate the nature of the relationship between recovery and employment hope.

Reviewer 2 Report

Comments and Suggestions for Authors

The paper covers an interesting topic and tries to fill the gap in research concerning the relationship between subjective aspects of employment and perceptions of global recovery for people with serious mental illness. The analysis was conducted on the basis of a 276-item sample limited to data collected in from March 2021 to January 2023 in from three in three midwestern US states. To evaluate constructs under consideration: employment hope, subjective recovery, psychiatric symptoms and mental health burden, the Authors use measurement scales described in the literature and characterized by other researches as having strong validity and reliability. Statistical tools used in the study include: descriptive measures, correlation analysis and hierarchical regression analysis. Despite numerous limitations (most participants male with a military background; subjective – self-reported approach, e.t.c.) that reduce the possibility of generalization of the results, the article is valuable work and presents interesting conclusions.

However, I have some remarks/doubts:

1.        The Authors use scales which are described in the literature as demonstrating strong validity and reliability but they do not mention if they checked aspects of validity/reliability of the scales in this particular study (e. g. Cronbach’s alpha, AVE, CR or other measures).

2.        Line 192: “All assumptions of statistical tests were met.” It needs explanation what the authors mean by “all assumptions” because it is unclear, e.g. when testing the significance of the correlation coefficient normal distribution comes to mind as an assumption…

3.        Lines 192-194. “Of note, multicollinearity among the independent variableswas not an issue after examining variance inflation factor and tolerance statistics.” The VIF measures and/or tolerance values should be provided and the accepted threshold should be given.

4.        Paragraph 3.3. Linear Regression Analyses. In my opinion the description is too compact. As I understand, the Authors use hierarchical regression analysis in the following way: in the first step (first block in stepwise entry in SPSS) they include CSSM symptom total scores and VR-12 MCS scores as independent variables; in the second step (second block in stepwise entry in SPSS) RAS variable is added. I hope, I am right because the description provided in the article is not clear enough. The Authors should underline how they performed regression analysis, why they decided to use stepwise procedure (I find it as an advantage of this study) and what conclusions can be drawn from this approach.

5.        Paragraph 3.3. Linear Regression Analyses. The Authors use two expressions: (1) EHS-14 mean scores (2) EHS-14 total scores. From the context I guess that these are the same variables and if so please unify because this ambiguity makes a bit difficult to understand the content.

Author Response

  1. The Authors use scales which are described in the literature as demonstrating strong validity and reliability but they do not mention if they checked aspects of validity/reliability of the scales in this particular study (e. g. Cronbach’s alpha, AVE, CR or other measures).

Response: We have further expanded this section; in particular, we have added cronbach's alpha for employment hope (EHS-14 total and EHS-14 subscale scores) and explaining why we did not check validity/reliability for other scales, particularly when psychometric properties have been substantially examined by a large number of prior studies in samples of participants with serious mental illness.

  1. Line 192: “All assumptions of statistical tests were met.” It needs explanation what the authors mean by “all assumptions” because it is unclear, e.g. when testing the significance of the correlation coefficient normal distribution comes to mind as an assumption.

Response: We added in add statistical assumptions that were met for this multiple regression analysis and how they were met (e.g., examination of residual plots)

  1. Lines 192-194. “Of note, multicollinearity among the independent variables was not an issue after examining variance inflation factor and tolerance statistics.” The VIF measures and/or tolerance values should be provided and the accepted threshold should be given.

Response: We have added in all VIF and tolerance statistics for all independent variables in the final linear regression model explaining employment hope. We have also included values and their thresholds that indicate multicollinearity may be an issue. In our study, those values indicate multicollinearity was not a problem.

  1. Paragraph 3.3. Linear Regression Analyses. In my opinion the description is too compact. As I understand, the Authors use hierarchical regression analysis in the following way: in the first step (first block in stepwise entry in SPSS) they include CSSM symptom total scores and VR-12 MCS scores as independent variables; in the second step (second block in stepwise entry in SPSS) RAS variable is added. I hope, I am right because the description provided in the article is not clear enough. The Authors should underline how they performed regression analysis, why they decided to use stepwise procedure (I find it as an advantage of this study) and what conclusions can be drawn from this approach.

Response: We agree that the description of our use of stepwise linear regression is too brief and does not indicate the strength of this approach. Therefore, we have expanded the discussion of this approach in the Data Analysis section (2.5) indicating how this allowed us to control for symptoms and high level of symptom burden on daily life while better honing in on the effects and association of employment hope on subjective recovery. In addition, we added in another portion explaining this approach in section 3.3.

  1. Paragraph 3.3. Linear Regression Analyses. The Authors use two expressions: (1) EHS-14 mean scores (2) EHS-14 total scores. From the context I guess that these are the same variables and if so please unify because this ambiguity makes a bit difficult to understand the content.

Response: Thank you for pointing out this issue. We have clarified such that both say "EHS-14 mean scores".

Reviewer 3 Report

Comments and Suggestions for Authors

Thank you for the opportunity to review this manuscript. This is an important area of study! While I found the theoretical/background strong (e.g. description of the recovery approach, info about SMI and employment, concepts of employment hope), I believe that an important piece is missing from this section. Since the study population is Veterans, some background about this population and SMI, employment, and other key concepts in this study should be covered. 

In the Results section, page 4 line 185, the numbers don't add up. 88% plus 22% = 110%.

Also, the CBTw intervention is mentioned in the methods section, but it is not clear how this is a component of this study. I am assuming that it is not, as it is selection criteria that was used for a different study. Perhaps the authors could add a sentence that states that the sample was created for an intervention study, reported elsewhere, etc. I do see that CBTw is brought back into the discussion section, but it just felt a little disconnected since the role of CBTw for this particular analysis was not addressed. 

Author Response

  1. Thank you for the opportunity to review this manuscript. This is an important area of study! While I found the theoretical/background strong (e.g. description of the recovery approach, info about SMI and employment, concepts of employment hope), I believe that an important piece is missing from this section. Since the study population is Veterans, some background about this population and SMI, employment, and other key concepts in this study should be covered.

Response: We agree that a section on veterans tying back to the literature is needed, therefore, we have added a section in the Introduction examining this and discussing veterans with SMI and employment, as well as the impacts of unemployment on various domains of recovery and well-being.

  1. In the Results section, page 4 line 185, the numbers don't add up. 88% plus 22% = 110%.

Response: We appreciate this reviewer identifying our error. We have corrected this to reflect that 78% identified as male and 22% identified as female or transgender female.

  1. Also, the CBTw intervention is mentioned in the methods section, but it is not clear how this is a component of this study. I am assuming that it is not, as it is selection criteria that was used for a different study. Perhaps the authors could add a sentence that states that the sample was created for an intervention study, reported elsewhere, etc. I do see that CBTw is brought back into the discussion section, but it just felt a little disconnected since the role of CBTw for this particular analysis was not addressed.

Response: We have further clarified that this study is a secondary analysis of baseline data from a clinical trial that tested CBTw vs. psychoeducation for unemployed veterans with serious mental illness.